



# Characterization of physical properties of a coastal upwelling filament with evidence of enhanced submesoscale activity and transition from balanced to unbalanced motions in the Benguela Upwelling Region

Ryan P. North[1,*], Julia Dräger-Dietel[1,*], and Alexa Griesel[1,*]

[1]Institute of Oceanography, University of Hamburg, Hamburg, Germany

[*]These authors contributed equally to this work.

**Correspondence:** Alexa Griesel alexa.griesel@uni-hamburg.de

**Abstract.** We combine high-resolution in-situ data (ADCP/Scanfish and surface drifters) and remote sensing to investigate the evolution, lifetime and physical characteristics of a major filament observed in the Benguela upwelling region. The 30-50 km wide and about 400 km long filament persisted for at least 40 days. Mixed layer depths were less than 40 m in the filament and over 60 m outside of it. Observations of the Rossby number Ro from the various platforms provide the spatial distribution of Ro for different resolutions. Remote sensing focuses on geostrophic motions of the region related to the mesoscale eddies that drive the filament formation and thereby reveals $|\text{Ro}| < 0.1$. Ship based measurements in the surface mixed layer reveal $0.5 < |\text{Ro}| < 1$, indicating the presence of unbalanced, ageostrophic motions. Timeseries of Ro from triplets of surface drifters trapped within the filament confirm these relatively large Ro and show a high variability along the filament. A scale dependent analysis of Ro, which relies on the $2^{nd}$ order velocity structure function, was applied to the latter drifter-group and to another drifter-group released in the upwelling zone. The two releases explored the area nearly distinctly and simultaneously, and reveal that at small scales ($< 15\,\text{km}$) Ro values are twice as large in the filament in comparison to its environment with $\text{Ro} > 1$ for scales smaller than $\sim 500\,\text{m}$. This suggests that filaments are hotspots of ageostrophic dynamics, pointing to the presence of a forward energy cascade. The different dynamics indicated by our Ro-analysis are confirmed by horizontal kinetic energy wavenumber spectra, which exhibit a power law $k^{-\alpha}$ with $\alpha \sim 5/3$ for wavelengths $2\pi/k$ smaller than a transition scale of $15\,\text{km}$, supporting significant submesoscale energy at scales smaller than the first baroclinic Rossby Radius ($\text{Ro}_1 \sim 30$ km). The detected transition scale is smaller than those found in regions with less mesoscale eddy energy, consistent with previous studies. We found evidence for the processes which drive the energy-transfer to turbulent scales. Positive Rossby



numbers $\mathcal{O}(1)$ associated with cyclonic motion inhibit the occurrence of positive Ertel Potential Vorticity and stabilize the water column. However, where the baroclinic component of EPV dominates, submesoscale instability analysis suggests that
mostly gravitational instabilities occur and that symmetric instabilities may be important at the filament edges.

**Short Summary**

The Benguela upwelling region off the coast of Namibia supplies cold water from the deep ocean that is transported offshore in finger-like structures called filaments. We investigate one major filament using measurements from a ship that crossed it multiple times and with mutiple buoys that follow the ocean currents. We find that the motions associated with the filament
enhance the kinetic energy at small scales and provide a pathway for mixing of water and turbulent dissipation of energy.

## 1   Introduction

The Benguela upwelling region is located off the coast of Southwest Africa and is one of four major Eastern boundary up-welling systems. Of these systems, the Benguela system exhibits some of the strongest upwelling cells and at the same time, in climate models, some of the largest errors in sea surface temperature (SST) (see Fig. 1 of Wang et al. (2015)). The Benguela
system consists of a northward flowing Benguela coastal current driving upwelling cells that vary in strength and persistence with latitude. One major upwelling cell occurs off Lüderitz (Namibia), and divides the Benguela upwelling system into a northern and southern section, where the latter is more affected by Agulhas rings that have become trapped in the Benguela current. The offshore edge of the system is defined by a coastal upwelling front, which is interrupted by filaments —finger-like structures that jut out from the boundary current, and are associated with strong offshore jets that carry cold water offshore
(Hösen et al., 2016). These mesoscale filaments play a key role in the exchanges of water masses between the frontal region of upwelling cells and the wider ocean basin. They are of great importance due to the intense biological productivity near-shore and their fertilizing role of the adjacent nutrient-deficient ocean (Álvarez-Salgado, 2007). Concerning climate models a better understanding of their meso- and submesoscale processes is needed in order to improve the parameterisations of their small scale dynamics. Previous studies of filaments from the Benguela upwelling region focused on overall statistics characterization,
namely seasonal variability, filament size, occurrence and lifetime statistics or circulation and offshore transport (e.g. Hösen et al., 2016; Kostianoy and Zatsepin, 1996; Muller et al., 2013; Shillington et al., 1992). Insight into a single filament, namely characterisation of the structure and cross-shore transport, could be gained by the calculation of three-dimensional finite size Lyaponov exponents by means of a realistic regional simulation of the western Iberian shelf (Bettencourt et al., 2017). While (Barton et al., 2001) found that the mixing within the core seems to be restricted to the upper layers, most recently, Peng et al.
(2020, 2021) showed that the fronts along the edge of filaments are regions susceptible to instabilities that lead to turbulence and energy dissipation. Desirable would be not only to improve understanding about the characterization of the filaments' physical characteristics and anomalies but also to compare them with the embedding environment.



In this paper, we focus on a major filament that was observed during a cruise in November/December 2016 with a particular emphasis on the characterization of the meso- to submesoscale transition linked to the filament in comparison to its environment. Through a combination of in-situ and remote sensing, we provide a detailed description of the filament characteristics and its evolution and lifetime, including the spatial distribution of geostrophic and ageostrophic motions from different platforms. The latter is mainly determined through variations in the Rossby number (Ro), by identifying regions with relatively high values, suggesting dominance of ageostrophic motions. The Rossby number describes the relative importance of inertial to Coriolis force and is defined as $U/fL$, where $U$ is velocity, $f$ is the coriolis frequency and $L$ is a length scale. It can be estimated from the vertical relative vorticity $\zeta$ by Ro$= \zeta f^{-1}$ and serves as a measure of the ratio of balanced and unbalanced motions. Ro is small ($\mathcal{O}(<0.1)$) for quasigeostrophic motions that feature an inverse energy cascade and hence point to kinetic energy depletion at small scales. Higher Ro values (e.g., Ro$> 0.5$) indicate that unbalanced, ageostrophic motions are present, and may drive a forward energy cascade, whose strength at a certain scale seems to depend on the presence of ageostrophic dynamics (e.g., Molemaker et al., 2010; Brüggemann and Eden, 2015). We investigate here, whether and where the filament is associated with significant submesoscale motions, defined here as motions occurring on a scale of $\mathcal{O}(10$ km$)$ —i.e., less than the first baroclinic Rossby Radius of deformation —and whether they are associated with small or large Rossby numbers. Through transects, drifters, and remote sensing, we obtain distributions of Ro in the horizontal and vertical, and at different spatial resolutions and temporal variations. Where high Ro are found, we aim to identify likely processes (instabilities) linked with a forward energy cascade by using Ertel Potential Vorticity (EPV) and Richdarson number (Ri). The combination of Lagrangian/Eulerian methods and a variety of platforms provides both a regional and local perspective with different resolutions. Namely, remote-sensing and full cruise data provide a regional perspective, while high-resolution data across (Scanfish/ADCP) and along (drifter) the filaments provide the local perspective.

In addition to Ro, ADCP and remote sensing data provide horizontal kinetic energy (KE) wavenumber spectra, whose slope $\alpha$ depends on the underlying dynamics. Quais-Geostrophic (QG) turbulence predicts a power law $k^{-\alpha}$ with $\alpha = 3$ for scales smaller than the forcing scale (generally associated with the wavelength of the baroclinically most unstable mode). In the presence of significant submesoscale energy, the spectra are expected to flatten for smaller scales. Lindborg (2005) explain the flatter spectral slope of $k^{-5/3}$ by a forward cascade due to non-linearly interacting gravity waves. Chereskin et al. (2019) find for the California upwelling region that KE is dominated by balanced motions for horizontal scales greater than 70 km and that at scales of 40-10 km unbalanced motions contribute as much as balanced ones. The transition in spectral slope from shallow to steeper slopes is missing for regions lacking an eddy field as energetic as in the Western Boundary currents. Instead, the scale at which the flow changes from balanced to unbalanced, and from a reverse to forward energy cascade, may be identified by a flattening of the KE spectral slope at submesoscales (e.g., Callies and Ferrari, 2013; Rocha et al., 2016; Chereskin et al., 2019; Sukhatme et al., 2020). This transition may also be identified by a change in the relative contribution of rotational and divergent velocity components (Chereskin et al., 2019), determined from across- and along-track velocities through the Helmholtz decomposition method of Bühler et al. (2014). The transition marks a shift to the dominance of ageostrophic over balanced geostrophic motions, and occurs at smaller and smaller scales as the region's mesoscale energy increases (Chereskin et al., 2019).



Here, we consider whether the Benguela upwelling region, with its enhanced eddy kinetic energy stemming from the passage of Agulhas rings (eddies), exhibits spectral characteristics that differ from the California upwelling region. In addition, the intention is to investigate whether the filament exhibits hotspots of unbalanced flow and how these hotspots vary over space (and to some extent, time), along or across the filament. We then consider whether filaments are pathways for transferring energy from mesoscale to submesoscale and turbulent scales and if so, at which scales? What is the spatial distribution/variability of this forward energy cascade in the filament? Can we identify/quantify the instabilities that are forming?

The paper is organized as follows: Section 2 describes the combination of methods (Lagrangian/Eulerian) and platforms: remote-sensing and full cruise data provide a large-scale regional perspective, while high-resolution data across (Scanfish/ADCP) and along (drifters) the filament provide a more local perspective. Section 3 describes the results in terms of filament formation, structure and evolution off-shore, temperature, salinity and velocity characteristics and evolution in time. Section 4 discusses further analyses to decipher the dynamical properties: Kinetic energy spectra to identify possible spectral slopes and transition scale, Rossby number distributions from SSH and ADCP transects as well as the along-filament distribution of Rossby numbers from drifters, and submesoscale instabilities. Section 5 provides discussion and conclusions.

## 2 Data and methods

The RV Meteor cruise took place between November 15th and December 11th, 2016, during the southern hemispheric spring/-summer, when the formation of filaments in the region west of Löderitz, Namibia ($28°$ to $24.5°$ S) is at a maximum (Hösen et al., 2016). The in-situ observations targeted the front along the western boundary of the upwelling zone, and a filament that was identified at the start of the cruise at a latitude of approximately $26.5°$ South (Figure 1). This study focuses on measurements taken within and around the filament with a ship mounted Acoustic Doppler Current Profiler (ADCP), a Scanfish (description below) towed by the ship, and autonomous drogued drifters. The ship crossed the filament nine times at four different locations. The first two crossings were completed while towing the Scanfish, and are a special focus of this study. The four longest transects were use to derive the kinetic energy spectra. The first drifters were deployed after the third filament crossing, along the southern boundary of the filament. The ship transects covered approximately 150 km of the filament length, and occurred over a period of 15 days. They therefore capture both, the spatial and temporal variability of the filament.

### 2.1 Ship-based measurements

The MS Meteor was equipped with a 75-kHz Ocean Surveyor ADCP, which, after processing, provided current velocity measurements at a vertical resolution of 8 m and a horizontal resolution of 1 km (300 s ensemble average bins). The Common Ocean Data Access System (CODAS) processing tools were used to clean, correct (e.g., for ship speed and heading and beam angle) and ensemble average the data. Before determining horizontal differentiation of zonal and meridional velocities (e.g., du/dx), the ship track was rotated to match the velocity components.

To obtain density information at a similar vertical and horizontal resolution to the ADCP, a Scanfish Mark II from EIVA a/s, Denmark was deployed. The Scanfish is a towed, undulating platform that can be controlled from the ship. The Scanfish was



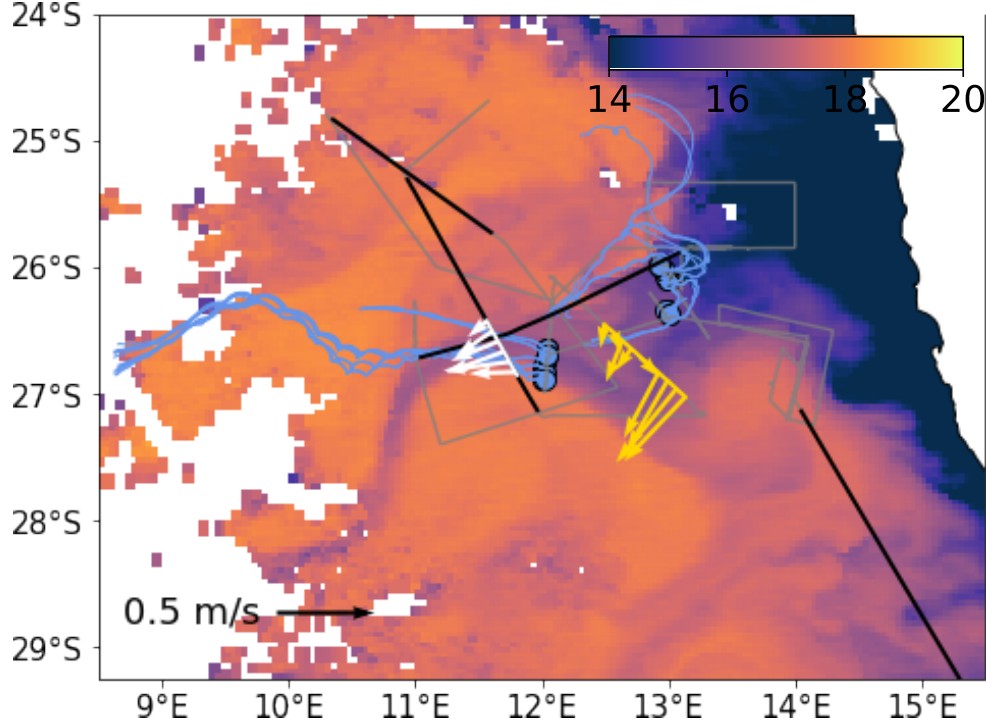

**Figure 1.** The study region off the coast of Namibia, showing sea surface temperature (SST) as color contours, the location of the nearshore (yellow line) and offshore (white line) Scanfish transects and the velocities (20 m below the surface) measured along these transects (coloured vectors). The ship track from the full cruise is shown as a thin grey line and the thick black lines show the transects used to determine the kinetic energy spectra. The SST data are from November 18, 2016. The blue lines represent the 14-day trajectory of the two separate drifter deployments (23 drifters), which are in the focus of this paper, and two drifter groups (12 drifters) deployed a 10 days respectively a week later at same locations.

equipped with conductivity, temperature and pressure sensors, among others. The Scanfish was set to undulate to a maximum

depth of 150 m, such that the maximum horizontal distance between two measurement points at a given depth was about 2

km. The data were linearly interpolated to uniform vertical and horizontal spacing. Because of the large difference between

the vertical and horizontal spacing, the interpolation was done in two steps: first along the instruments flight path, and then

horizontally to create uniform gridded data (North et al., 2016). A horizontal spacing of 1 km and a vertical spacing of 4 m was

chosen to match the ADCP data, while preserving the higher vertical resolution of the Scanfish. The ADCP data was therefore

linearly interpolated to a vertical spacing of 4 m when used in conjunction with the Scanfish data.

The two Scanfish transects that crossed the filament occurred in succession over a period of 44 hours, with each crossing

taking 10-13 hours. The transects are approximately 100 km apart, with the first crossing occurring at a narrow section with the

strongest horizontal SST gradients (thick white line in Figure 1), and the second crossing occurring further inshore where the

filament was at its widest (thick yellow line in Figure 1). The first transect will be referred to as the offshore Scanfish transect,

and the second as the nearshore Scanfish transect.





All conductivity and temperature measurements were converted to absolute salinity ($S_A$) and conservative temperature ($\Theta$) (IOC et al., 2010), from which potential density ($\rho_\theta$) was calculated. For simplicity, these three variables will simply be referred to as salinity, temperature and density in the text. The analysis mainly focuses on the surface mixed layer (SML), which was

defined as sitting above the isopycnal that exceeded the surface density by 0.1 kg m$^{-3}$. The bottom of the SML generally aligns with a maximum in the depth dependent buoyancy frequency ($N(z)$), where $N(z) = \sqrt{(-g\,\rho^{-1})(\partial\rho_\theta\,\partial z^{-1})}$, $g$ is the acceleration due to gravity, $\rho_\theta$ is a reference density (1000 kg m$^{-3}$) and $z$ is the depth below the surface.

The wavenumber spectral analysis will focus on kinetic energy (KE$= 0.5(u^2 + v^2)$) derived from zonal ($u$) and meridional ($v$) velocity components measured between 25 m and 50 m below the surface (approximate depth of the SML) along the

four longest transects (thick black lines in Figure 1). These transects provide the widest coverage of wavenumbers —due to their lengths— and regions or features of interest —due to their spatial coverage. The first transect (160 km long) crossed the northwestern side of the mesoscale eddy that was located on the northern side of the filament. The second transect (227 km long) also crossed this mesoscale eddy, and includes the first Scanfish section that crossed the filament (thick white line in Figure 1). The third transect (160 km long) followed the northern edge of the filament, from the upwelling region to 11°

East. The fourth transect (273 km long) was obtained as the ship left the study area, on route to Capetown, South Africa, and generally passes through the upwelling region or its western edge.

**Method: Kriging the ADCP data**

The $u$ and $v$ velocity components measured nearest to the surface by the ADCP, were spatially interpolated across the study area —i.e., the region covered by the ship's track. The resulting interpolated and gridded velocities could then be used to estimate

the spatial variability of Ro. As velocity measurements were unevenly distributed across the study area, interpolation was performed using two-dimensional Ordinary Kriging, also referred to as Gaussian process regression. Kriging first determines the spatial autocorrelation between measurement points. A model is then fit to the autocorrelation data to determine a relationship between distance and direction between points. The resulting modelled fit can then be used when filling in gaps not covered by the ship's measurements. This means that the velocity at a point where the ship did not pass is determined from nearby

velocity measurements. However, instead of being influenced by just the closest measurements (as in linear interpolation), depending on the model that is chosen, any number of velocity measurements can influence the estimated velocity. For example, when using the spherical model, there is a gradual decline in the influence (i.e., in autocorrelation) of neighbouring velocity measurements with distance, before a cutoff distance is reached and the influence drops to zero. Kriging was performed using the Python toolkit PyKrige (geostat-framework.readthedocs.io/projects/pykrige/en/stable/), and the optimal data setup, model

and fit were determined by tuning the parameters. In particular, the tuning determined that the best fit was obtained with a spherical variogram model and the ADCP data set of the full cruise, after removing measurements taken when the ship's speed was below 2.1 m/s (i.e., when slowed or stopped to take additional measurements). For ease of processing, the ADCP data was reduced to 30 minute intervals, which did not affect the accuracy of the result.



## 2.2 Drifters

The two drifter releases that are used in this paper were deployed during the early stage of the filament in two separate regions. In the following we will refer to the two releases as the southern and northern releases, where the southern release was deployed near $12°3'$E, $26°40'$S and the northern release near $12°58'$E, $26°3'$S (Figure 1). The northern release was deployed within the upwelling region and the southern release was deployed at the southern edges of the filament. Each release consist of four clusters of three satellite-tracked surface drifters. Each cluster formed a triangle, whose sides were approximately 200 m long.

The clusters were deployed 5 km apart, resulting in a time-delay of 1 hour between deployments. For details of the drifter deployment strategy see Dräger-Dietel et al. (2018), which focused on temporal evolution of the relative dispersion of drifter pairs and performed a finite size Lyaponov-analysis in the region.

The drifters were of type SVP-I-XDGS from metOcean, and included a satellite transmitter (Iridium) mounted inside a $16''$ surface buoy, a surface temperature sensor, and a drogue centered in the mixed layer at a depth of 15 m to estimate ocean

currents at this depth level. According to the manufacturer's specifications, the GPS position errors are smaller than 2.2 m with a 50% confidence level, and smaller than 5.5 m with a 98% confidence level (GPS receiver Jupiter 32, Navman, UK). For normally distributed errors, this implies a standard deviation of 1.86 m. Data transmission was set to 30 minute intervals from the date of deployment (02 December 2016) until 06 April 2017. After this date, the operation of the devices was transferred to the Global Drifter Program, setting the temporal resolution to 6 hours.

We used the trajectories of the drifter triplets to investigate the Lagrangian evolution of the Rossby number (from relative vorticity), where we applied two methods described in detail in Molinari and Kirwan (1975) to test the robustness of the results. The first method is based on a least squares fit of the observed drifter velocities to estimate horizontal velocity gradients ("the linear least squares method"). The second method traces back to a method of Saucier (1955) for computing atmospheric kinematic properties and was applied to ocean drifter data first in Molinari and Kirwan (1975). This method makes use of the

fact that the horizontal divergence can be expressed as the fractional time rate of changes of the horizontal area A of a parcel, here a triad of drifters. According to Saucier (1955) vorticity can be evaluated by $90°$ clockwise rotation of the velocity vectors of the three drifters ("the triangle method"). Similar to Essink et al. (2022) we found the triangle method to be more robust (i.e. less sensitive when the aspect ratio of the triangles becomes too large) than the linear least squares method, and we only show the results using this method here.

While the drifters of the southern release generally remain trapped in the filament, drifters from the northern release followed many different currents across a larger region. The unique situation, that the two releases explored nearly distinct but nearby areas, enables us to explore the filament and its environment quasi separately and simultaneously. Namely the two releases could be used to perform a scale dependent analysis of the Rossby number, which reflects the amount of submesoscale activity within and outside the filament. The method relies on the $2^{nd}$ order velocity structure function and thereby treats the drifter

velocities as scattered (Eulerian) measurements (Balwada et al., 2016; Poje et al., 2017) as described in more detail in section 4.2.4.



## 2.3 Remote sensing

Daily MODIS-Aqua Level-3 mapped sea surface temperature (SST) data (ca. 4 km resolution) covering the full cruise period were obtained through the Ocean Biology Processing Group's website OceanColor Web (OBPG, 2018). Concurrent
SSALTO/DUACS Delayed-Time Level-4 sea surface height (SSH) derived variables (0.25° resolution) were obtained through the Copernicus Climate Data Store (https://cds.climate.copernicus.eu/, 2019). These variables include sea level anomaly (SLA) — the sea surface height above mean sea surface (referenced to 1993-2012) — and absolute geostrophic velocity.

## 3 Results

### 3.1 Filament formation and duration

During the study period, mesoscale eddies with Rossby numbers (Ro; 0.2) that exceeded the regional average were present in the study area, along the western coast of South Africa and Namibia. A time-series of SLA images (not shown) showed the eddies advecting northwards from the southern tip of the African continent, suggesting they originated in the Aghulas current. Over the course of the study, winds were consistently northwards, and ranged from less than 5 m s$^{-1}$ to more than 15 m s$^{-1}$, but generally hovered around 10 m s$^{-1}$. The winds drove offshore transport of coastal surface waters through Ekman transport,
resulting in the upwelling of colder deep water, visible in SST images of the study region (Figure 1).



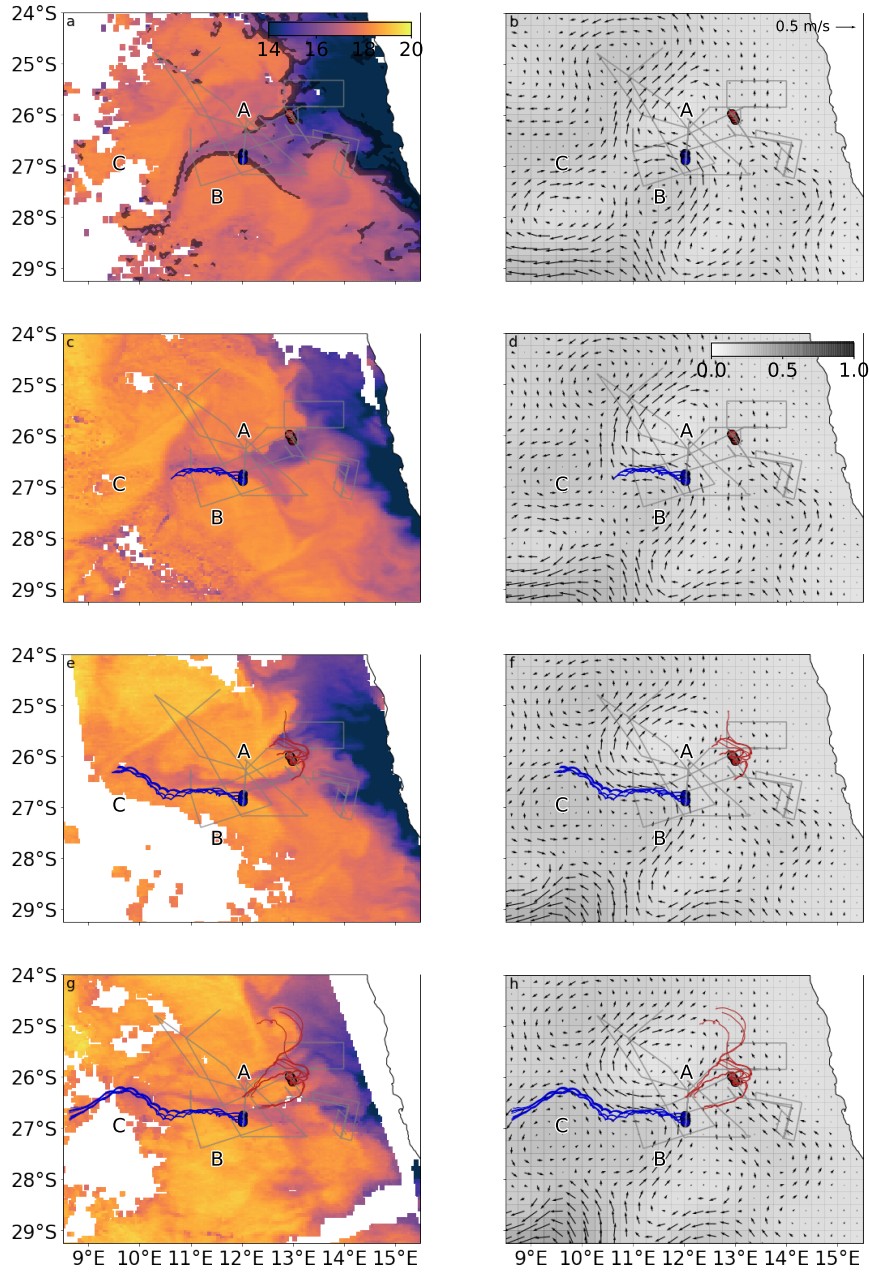

**Figure 2.** Sequence of mapped sea surface temperature (SST) images (left panels) and sea level anomaly (SLA) (right panels), showing the development of the filament during the study period. Image dates correspond to a/b) 2016-11-18, c/d) 2016-11-25, d/f) 2016-12-01, and g/h) 2016-12-05. SLA derived geostrophic velocity estimates are plotted as black arrows (right panels). Overlaying the images are the drifter deployment positions (filled circles) and tracks up to the date of the respective SST/SLA image. Drifter trajectories from the first (southern) deployment are shown in blue, while drifter trajectories from the second (northern) deployment are shown in red. The ship track from the full cruise is shown as a thin grey line. The letters ('A','B' and 'C') mark the approximate position of select mesoscale eddies, identified from SLA and resulting geostrophic velocities. In (a), the grey shading indicates estimated horizontal temperature gradients that exceed 0.03 °C km$^{-1}$. The image dates were selected based on the quality of the SST image and to cover as much of the study period as possible.





The upwelling produced a temperature difference of as much as 5° C between coastal and offshore waters (Figure 2). The edge of the upwelling region was defined by an $\mathcal{O}(10)$ km wide front with horizontal gradients of at least 0.03° C km$^{-1}$ (dark grey shading in Figure 2a). The upwelling region extended from 50 km to 200 km offshore and remained for the full study period, despite a gradual increase in SST in the ambient offshore water and the filament (Figure 2).

At the start of the cruise, SLA images and geostrophic velocities show a pair of counter-rotating eddies on the western side of the upwelling front ('A' and 'B' in Figure 2b). The surface current generated between the eddy pair was directed offshore, drawing water from the upwelling region and forming a filament of relatively cold, fresh and dense water. On November 18th, the 3rd day of the cruise, the filament was visible in a SST image as a winding strand of relatively cold water starting at the upwelling front (approximately 200 km offshore) and extending southwestward over more than 400 km (Figure 2a). The
filament was bounded by horizontal temperature gradients on the order of 0.03° C km$^{-1}$ (dark grey shading in Figure 2a).

On November 21st, the first 12 surface drifters were deployed on the southern edge of the filament (Figure 2a). By combining a sequence of SST and SLA images with drifter tracks, the sinuous path of the filament becomes visible (Figure 2). The drifter tracks show how the filament wound its way between eddy pairs (not always counter-rotating) over hundreds of kilometres (Figure 2). The filament seemed to originate between an anticyclonic eddy to the north ('A' in Figure 2) and a cyclonic eddy
to the south ('B' in Figure 2). The meandering of the filament was due to the influence of successive eddies, in particular the apparent merging, weakening, strengthening and advection (mainly northward) of these eddies. For example, at first the SST shows the cold water filament turning southward at around 11 deg E (Figure 2a). However, by the time the drifters have reached this point, the filament has shifted northward, at around 10.5 deg E (Figure 2c). The geostrophic currents from SLA data suggest that this shift was driven by the appearance, or strengthening, of a cyclonic eddy ('C' in Figure 2f).

By the end of the cruise, a coherent filament path of relatively cool water was still visible in SST, but the horizontal gradients at the filament edges were greatly reduced (Figure 2g). The weak gradients may indicate the filament is reaching the end of its lifetime, as filaments in this region do not tend to last longer than 48 days (Hösen et al., 2016). However multiple drifters releases 10 days later (Figure 1) were still passing westward along a similar path as the original filament (not shown). They were apparently still being pushed by the anticyclonic and cyclonic eddies ('A' and 'C', respectively, in Figure 2). Therefore,
although the surface signature of the filament in SST may have disappeared, it would appear that the filament is still transporting water offshore.

### 3.2   Filament description

The two Scanfish sections show the filament was on the order of 2° C colder, 0.3 g kg$^{-1}$ fresher, and 0.3 kg m$^{-3}$ denser than the ambient water (Figure 3a and b). The SML is less than 40 m deep within the filament, and over 60 m deep outside of
the filament. At the bottom of the SML is a strongly stratified pycnocline, where $N^2$ is $\mathcal{O}(10^{-3})$ s$^{-2}$ (Figure 3e and f). The pycnocline is thinner on the southern side of the filament ($< 10m$) than the northern side ($> 20m$), and thinnest and weakest within the filament. When crossing the filament, the depth of the pycnocline increases towards the south. Below the pycnocline, the water is weakly stratified to depths below 100 m.





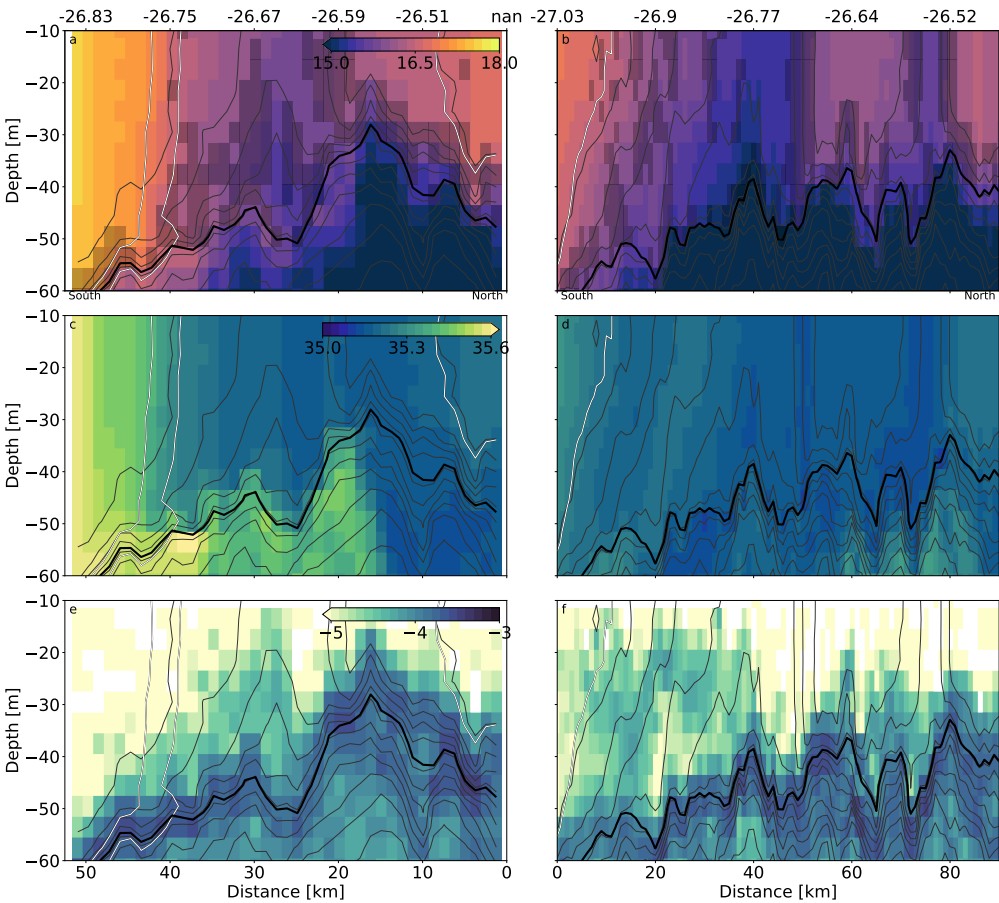

**Figure 3.** Vertical sections for the two Scanfish/ADCP transects that crossed the filament. The left (right) column of plots corresponds to the offshore (nearshore) transect (see Figure 2a). a/b) Conservative temperature, $T_C$, c/d) Absolute salinity, $S_A$, and e/f) buoyancy frequency, $\log(N^2)$. The thin grey contour lines show density (1024.2-1027.2 kg m$^{-3}$ in 0.05 kg m$^{-3}$ increments) and the thick black contour lines mark the depth of the surface mixed layer (SML) defined by a density change of 0.1 kg m$^{-3}$. The transects have been cropped (width and depth) to focus on the filament and the SML. The light grey shading in (a/b) indicates estimated horizontal buoyancy gradients that exceed $10^{-7}$ s$^{-2}$. The white contour lines indicate $T_C$ of 16.5∘C and 17∘C.

South of the filament, compensating lateral gradients in temperature and salinity (Figures 3a/b and 3c/d, respectively) re-
sulted in weak lateral buoyancy gradients. The southern front bounding the filament is on the order of 10 km wide, with sloping isopycnals — shallowest in the nearshore filament. The maximum lateral buoyancy gradients are found on the filament side of the front in the offshore transect, and towards the outside of the front in the nearshore transect (shading in Figure 3a and b). A stronger lateral buoyancy gradient ($>10^{-7}$ s$^{-2}$) was present on the north side of the filament in both transects, where isopycnals were nearly vertical. In the SML, salinity measurements show the fresher but denser filament water being pushed
southward across the southern edge of the filament (Figure 3c). Although this is clearer for Transect 1 (Figure 3c) than for Transect 2 (Figure 3d).



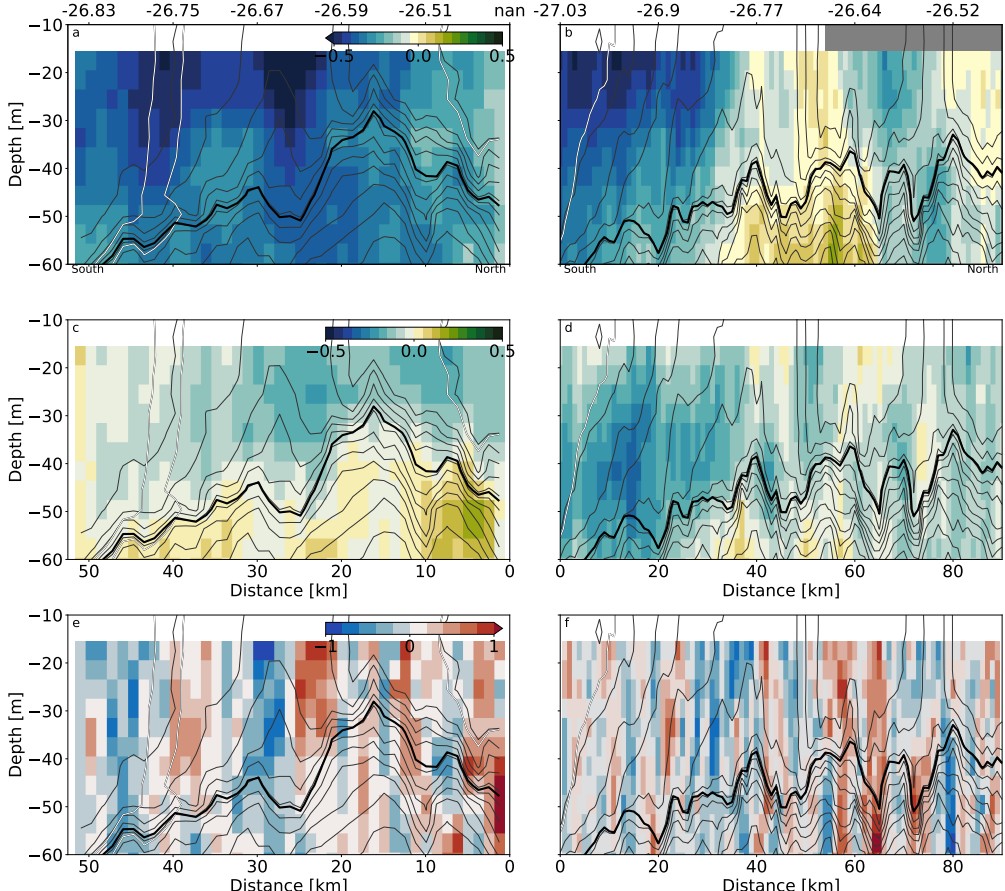

**Figure 4.** As in Figure 3 for a/b) along-filament velocity ($u$), c/d) across-filament velocity ($v$), and e/f) corresponding Rossby Number, estimated from the across-filament gradient of the along-filament velocity. The along- and across filament velocities correspond respectively to zonal and meridional velocity components for the offshore transect (a, c) and to across- and along-track velocity components for the nearshore transect (b, d).

The filament extends from approximately 26.6°S to 26.72°S (Figure 3a) at the offshore Scanfish transect, based on lateral buoyancy and temperature gradient maxima. The limits are not as clear in the nearshore transect, which passes through several patches of filament, each bounded by strong lateral buoyancy gradients (Figure 3b). The largest patch lies between 26.7°S

to 26.9°S, and is characterized by the relatively cold temperature, while a smaller patch to the north (26.6°S to 26.55°S) is warmer but saltier, resulting in similar densities for the two patches. The smaller patch is bounded on both sides by very strong and near-vertical lateral buoyancy gradients, similar to the northern boundary of the largest patch and of the filament in the offshore transect. Note that for the frontal analysis in Section 4.4, we focus on the largest filament.

In the offshore Scanfish transect (white transect in Figure 1), the along-filament velocity component can be approximated

by the zonal velocity ($u$), and the across-filament component by the meridional velocity ($v$). For the nearshore transect (yellow transect in Figure 1), the orientation of the filament and the angle of the transect meant that the across- and along-ship velocities





components align well with the the along- and across-filament directions, respectively. For both transects, the along-filament velocities were surface concentrated and confined to the SML. The strongest velocities (0.5 m/s) were found on the southern boundary of the filament, on both sides of the maximum in lateral buoyancy gradient maximum (Figure 4a and b), and in the

middle of the filament in the offshore transect (4a). In the northern front crossed by the nearshore transect velocities reversed, flowing eastward towards the coast.

In the SML, the maximum across-front velocities ($< 0.25$ m/s) were much lower than the along-front component and almost entirely southward (negative). In the offshore transect, the highest velocities were found in the filament core and in the middle of the southern front, with little variability in the vertical. The nearshore transect had higher values at both fronts and in the

filament core. However, the highest values were found close to the bottom of the SML in the southern front. Although the variability in across-front velocities suggest some convergence on the inside edge of the filament fronts, this is not particularly clear from the data. Convergence was however apparent in the tracks of the drifters that were deployed along the southern edge of the filament (blue lines Figure 2). The drifters did not spread apart as they were transported offshore (westward), suggesting they were trapped along the southern boundary of the filament, consistent with a region of convergence.

Distributions of velocity magnitude from altimetry based geostrophic velocity (over 30 days), drifters (30 days) and ship ADCP (23 days) indicate that the filaments are responsible for some of the strongest currents in the region. The drifter velocities, which remain trapped within the filament, show a peak in their distribution at approximately 0.3 m/s. Geostrophic velocity estimates do not capture these high velocities, but remain closer to 0.1 m/s, with only the tails of the distribution exceeding 0.3 m/s. Although the ship-based ADCP measurements, which cross the filament eight times, peak at less than 0.2

m/s, a second smaller peak extending beyond 0.3 m/s likely represents the contribution from the filaments.

## 4    Analysis

### 4.1    Kinetic Energy Spectra

Kinetic energy from the ADCP measurements in the SML were analysed to try and identify the scale of a possible transition from balanced to unbalanced flow for the study region. To reduce uncertainty, and to provide a more regional perspective,

the KE spectral analysis uses the four longest transects obtained during the cruise (thick black lines in Figure 1). The spectral analysis followed the methods of Rocha et al. (2016) and Chereskin et al. (2019), and used the Python package XRFT (https://github.com/xgcm/xrft). The spectral estimates were determined after decomposing the data into along- and across-track components and multiplying by a Hanning window. Spectra were determined at four depth levels (8 m bins) spanning 25 m to 50 m, and then averaged together. The selected depth range corresponds to the 2nd shallowest ADCP bin down

to the approximate bottom of the SML. Once averaged together, 95% confidence limits were determined before the spectra were smoothed by binning along the wavenumber axis. Finally, the depth-averaged spectra from the four transects were also averaged together to increase confidence in identifying and interpreting any transitions in the spectral slope.

The long transects that crossed one of the mesoscale eddies driving the filament's westward velocity, had the highest energy and steepest slopes ($\leq -3$) at large scales ($> 30$ km), and matching $-2$ slopes for scales between 30 km and 5 km. The other



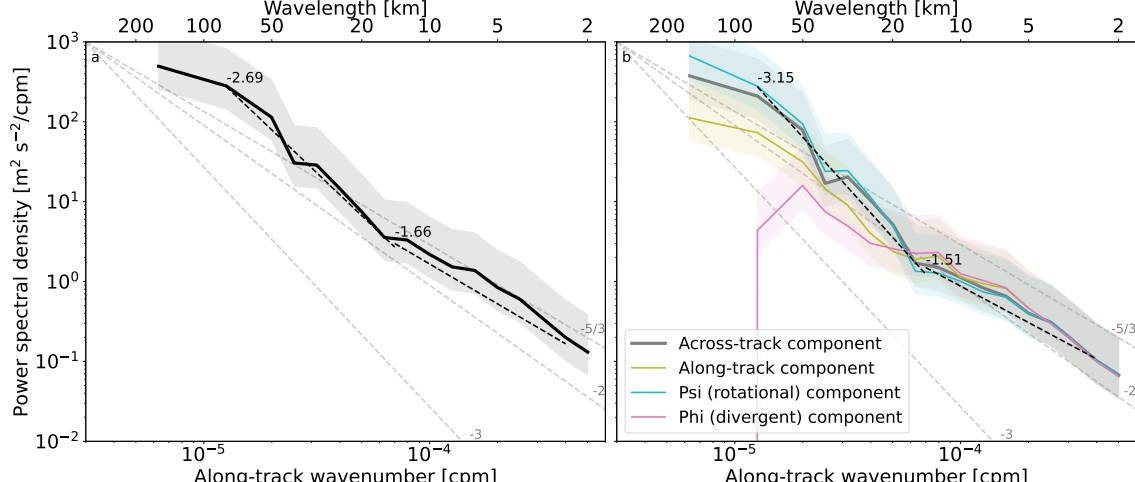

**Figure 5.** a) Depth-averaged (25 m to 50 m) kinetic energy (KE) spectra from the ship-mounted acoustic Doppler Current Profiler (ADCP) as a mean for the four long transects (black solid lines in Fig. 1).b) The mean KE spectra in (a) decomposed into its across- and along-track components and rotational- and divergent-components (Helmholtz decomposition). All plots include reference slopes of -5/3, -2 and -3 (dashed grey lines) and 95% confidence intervals (shading). The dashed black lines and text show slopes fitted to the mean spectrum in (a) and the Psi component in (b)

two long transects both had nearly uniform slopes across all scales, although the transect that followed the filament, had higher energy and a steeper slope ($-2$) than the transect that passed through the upwelling region (not shown). When the four spectra are averaged together, (thick black line in Figure 5a) they can be divided into three regimes : 1) a shallow slope from the largest wavelengths to just below 100 km reflecting the finite size of the transects (the longest transect was 270km long, see section 2.1); 2) the steepest slope ($-2.7$) down to a wavelength just below 15 km; and 3) a relatively shallow slope ($-1.66$) down to the smallest available wavelengths set by the resolution of the ADCP data (1 km).

The change in slope is emphasized when the average KE spectra is decomposed into its components (Figure 5b). At large wavelengths the contribution from the across-track component is several orders of magnitude above the along-track contribution (dark and light blue lines in Figure 5b). The contributions of the two components gradually converge with decreasing wavelength. At a wavelength of 15 km, the energy contained in the along-track component is comparable to that of the across-track. The transition at 15 km is also visible in the change in the relative contribution of rotational and divergent velocity components (gray and light red lines in Figure 5b). At larger wavelengths, the rotational component is more than an order of magnitude larger, but at wavelengths below 15 km, the divergent component contains more energy. The reversal at 15 km indicates that the larger scales are dominated by balanced flow (geostrophic), and unbalanced flow (ageostrophic) dominates at scales below 15 km. The change in slope and change in contribution to the spectra could indicate that a reverse energy cascade dominates at scales above 15 km, and a forward energy cascade at smaller scales below this threshold.









### 4.2  Rossby Number

#### 4.2.1  Regional near-surface variability in Ro

Rossby numbers Ro derived from the SLA data provide an overview of the spatial distribution of Ro across the study area (Figure 6). Ro was generally small ($|Ro| < 0.1$) in the upwelling zone and along most of its front. The highest Ro values
($|Ro| < 0.2$) were linked with mesoscale eddies, such as the cyclonic (anti-cyclonic) eddy to the north (south) of the filament (marked by "A" ("B") in Figure 2). Higher Ro values were also found south of the filament along the offshore edge of the upwelling zone. The filament itself generally showed low Ro ($|Ro| < 0.1$) with negative values on its southern side and positive values on its northern side. Aside from the movement of the eddies and the filament, the SLA data suggest that the large scale spatial variability of Ro remained constant over the course of the study period (Figure 6a to c).

Mapped values of Ro derived from ADCP current speeds (Figure 6d) had a similar spatial distribution to SLA derived Ro, despite the fact that the ADCP data provides a time-averaged map of Ro. This is not unexpected, given that the SLA derived snapshots of Ro changed very little with time. However, Ro from the ADCP were generally greater in magnitude than the corresponding SLA values. $|Ro|$ is up to 0.4 in the eddies and along the front of the upwelling zone, and exceeds 0.5 along the filament. Where the ship crossed the filament, the variability of Ro across the filament was better resolved by the ADCP data,
and, similar to the SLA data, shows positive values to the north and negative values to the south. This indicates a dominance of cyclonic motion on the northern side, and anticyclonic on the southern side, which is consistent with velocity gradients seen in the transects (e.g., see offshore Scanfish transect in Figure 1, white arrows). The variability of Ro along the filament was highly variable, which may be partly due to the lack of ADCP data between transects that crossed the filament. The higher Ro associated with the filament indicates that they may be dominated by ageostrophic motions and a forward energy cascade.
With frontal scales on the order of 15 km or smaller, this corresponds with the results of the spectral analysis.

Drifter triplets provide another source of Ro values. The drifters offer the possibility to infer velocity curl on smaller scales than the ship ADCP is able to resolve, since the drifters come as close as a few hundred meters to each other. Ro values were determined using the triangle method inferred from drifter triplets deployed in both the southern and northern releases that appear in the area of interest, over a period of 45 days. The resulting drifter derived $|Ro|$ (Figure 6e) are variable, but show
values that exceed 0.5. Based on the comparatively low Ro values of the few drifter triplets that do escape the filament or frontal regions, the drifter derived values also indicate that the upwelling fronts and the filament are regions of elevated Ro values.

#### 4.2.2  Cross-filament variability in Ro

Variability in Ro with depth and across the filament was provided by the ship's ADCP for two transects that included Scanfish
data (i.e., density and current velocities). The ADCP data only provides data along one path, and therefore the relative vorticity is estimated from $du/dy$, i.e., the across-filament gradient ($y$) of the along-filament velocity ($u$). This assumption is applicable for the transect data, because the magnitude of the along-filament velocities is much greater than the magnitude of the across-filament velocities. Ro values were consistently higher in the SML than below the pycnocline, and within the SML horizontal



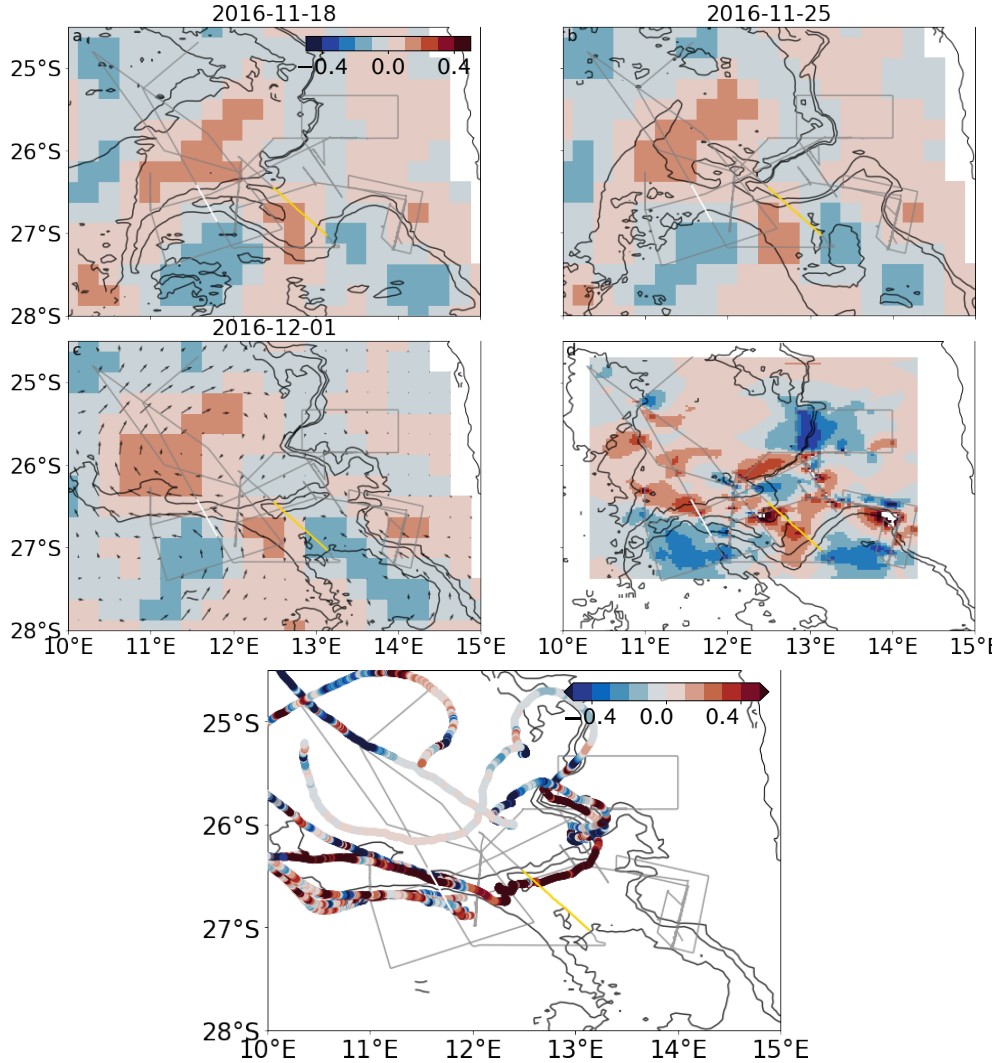

**Figure 6.** Top: Maps of Rossby number (Ro) derived from a/b/c) sea level anomaly (SLA), and d) the ship's ADCP at a depth of 17 m. The SLA results cover three different dates (see title) and the ADCP-based results are an average over the full expedition. The image dates were selected based on the quality of the SST and to cover as much of the study period as possible. Bottom: Rossby number as derived from 45 days of drifter trajectories with the triangle method with a 24 point Hanning filter using the triplets deployed both in the southern and northern deployment that travelled into the study region shown at different times. Overlaying all images are contours of sea surface temperature (16.5°C, 17°C, 18°C), the ship track (grey lines), and the offshore (white line) and nearshore (yellow line) Scanfish transects and, in (c), SLA derived geostrophic velocity estimates (black arrows).

variability in Ro was much more apparent than any vertical variability (Figures 4e and f). The core of the filament was
generally divided into negative Ro values to the south and positive to the north. For both transects, the core of negative Ro values decreased in magnitude as it crossed into the southern front, before reversing sign to become positive for 5 km of the outer edge of the frontal region. The sign of Ro eventually reversed again, becoming negative to the south of the filament, which



is consistent with the Ro maps (Figure 6). A similar, but weaker pattern of variability in Ro was found across the northern front, resulting in negative Ro values within the northern front, and positive values to the north of the filament; consistent once again with the maps of Ro. In general, there was very little variability of Ro in the vertical (Figures 4e and f). However, closer inspection of the southern frontal region suggests that the changes in sign discussed above, may follow the isopycnals; i.e., the positive or negative regions were not strictly uniform in the vertical, but along sloping isopycnals. However, the data is too coarse to determine this with certainty.

Although the magnitude of Ro varied across the transect and with depth, Ro was most consistently approaching 1 or -1 within the core of the filament, where horizontal velocity shear was the greatest. The general pattern of variability shown by the maps of Ro is consistent with the transect data, except that the increase in resolution shows higher variability across the frontal region.

### 4.2.3 Scale dependent Rossby number

By comparing the southern (within the filament) and northern drifter releases (within the upwelling region) using a scale dependent analysis of Ro, we can gain information on the regional sub-surface variability of Ro. This alternative view of the Rossby number relies on the second-order structure function, $S_2 = \langle \Delta w_\ell^2 \rangle + \langle \Delta w_t^2 \rangle$, where $\Delta w_\ell$ and $\Delta w_t$ are the longitudinal and transverse velocity differences between an arbitrary pair of drifters separated by the distance $r$, and $\langle \cdots \rangle$ denotes the average over a large number of drifter pairs conditioned on binned separation distances. More explicitly for each time step we calculated the longitudinal and transverse velocity increments

$$\Delta w_{\ell(i,j)} = (\boldsymbol{w}_i - \boldsymbol{w}_j) \cdot \frac{\boldsymbol{r}_{(i,j)}}{\|\boldsymbol{r}_{(i,j)}\|} \ , \quad \Delta w_{t(i,j)} = (\boldsymbol{w}_i - \boldsymbol{w}_j) \times \frac{\boldsymbol{r}_{(i,j)}}{\|\boldsymbol{r}_{(i,j)}\|} \ , \tag{1}$$

where $\boldsymbol{w}_i = (U_i, V_i)$ denotes the velocity of the $i$th drifter, and $\boldsymbol{r}_{(i,j)} = \boldsymbol{r}_i - \boldsymbol{r}_j$ the relative separation vector of drifter pair $(i,j)$ computed from the drifter position vectors $\boldsymbol{r}_i = (X_i, Y_i)$. Note that this approach treats the discrete velocities of a group of $N$ drifters as scattered-point Eulerian measurements, thereby ignoring potential biases with respect to convergent flow structures for small scales as described in Pearson et al. (2019). Following Balwada et al. (2016), we here use the second-order structure function $S_2$ to compute a scale-dependent version of the Rossby number: $|\text{Ro}| = \sqrt{S_2}/(fr)$.

The scale dependent analysis was applied to the two drifter-releases for over 130 days. Figure 7 shows that for small separation distances ($\mathcal{O}(<500\,\text{m})$), $|\text{Ro}|$ is $\mathcal{O}(1)$ for the southern release —twice that of the northern release —pointing to the presence of ageostrophic submesoscale dynamics. Because the contributions to this range of small separation distances are mainly from the initial period immediately after the drifter deployments in the filaments, this supports the idea that filaments are hotspots of forward energy cascade. At larger separation distances, mainly corresponding to periods after the drifters left the filament, the differences between the Ro-curves in Figure 7 diminish and yield $|\text{Ro}| \ll 1$, consistent with the mesoscale dynamics of the open ocean.





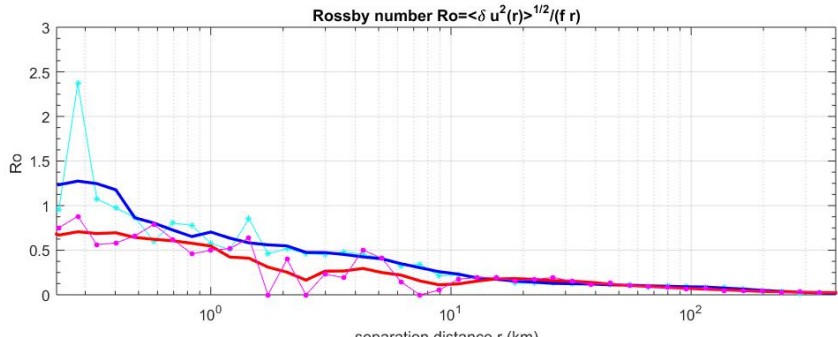

**Figure 7.** Scale-dependent Rossby number, $|\mathrm{Ro}| = \sqrt{S_2}/(fr)$, as a function of the binned separation distance r between arbitrary drifter pairs, for two releases. The southern drifter release (light blue curve with smoothed blue curve), which follows the filament (blue trajectories in Fig. 2). And the northern release (pink and red curve), where the drifters separate at an early stage and follow distinct currents of the upwelling front (red trajectories in Fig. 2). The data starts at deployment on 21 November 2016 (southern release) and 26 November 2016 (northern release) an extends until 06 April 2017 (136 days and 131 days, respectively).

## 4.3 Submesoscale instabilities

The in-situ measurements show that Ro values were sometimes of $\mathcal{O}(1)$ in the filament, which suggests that ageostrophic

motions become important and there is likely a forward energy cascade. But what evidence is there of processes that are driving this energy transfer to turbulent scales? The Richardson number Ri —the ratio between the vertical buoyancy gradient and the square of the vertical shear of horizontal velocity —is another key parameter to distinguish between different kinds of turbulent regimes. Large Ri $\gg 1$ indicate that the flow is mainly in quasi-geostrophic balance, while more ageostrophic effects feature smaller Ri (see e.g. Stone, 1966). By combining density observations from the Scanfish, with velocity from the ship's

ADCP, the gradient Richardson number $Ri^G = N^2(\partial v\, \partial z^{-1})^{-2}$ was calculated across the filament (Figure 8a-d). In order to establish across- and along-front axes (consistent with EPV and instability analysis), transects were split in order to focus on each front (north and south) separately. The axes were rotated so that the y-axis was in the along-front direction and positive in the downstream direction.





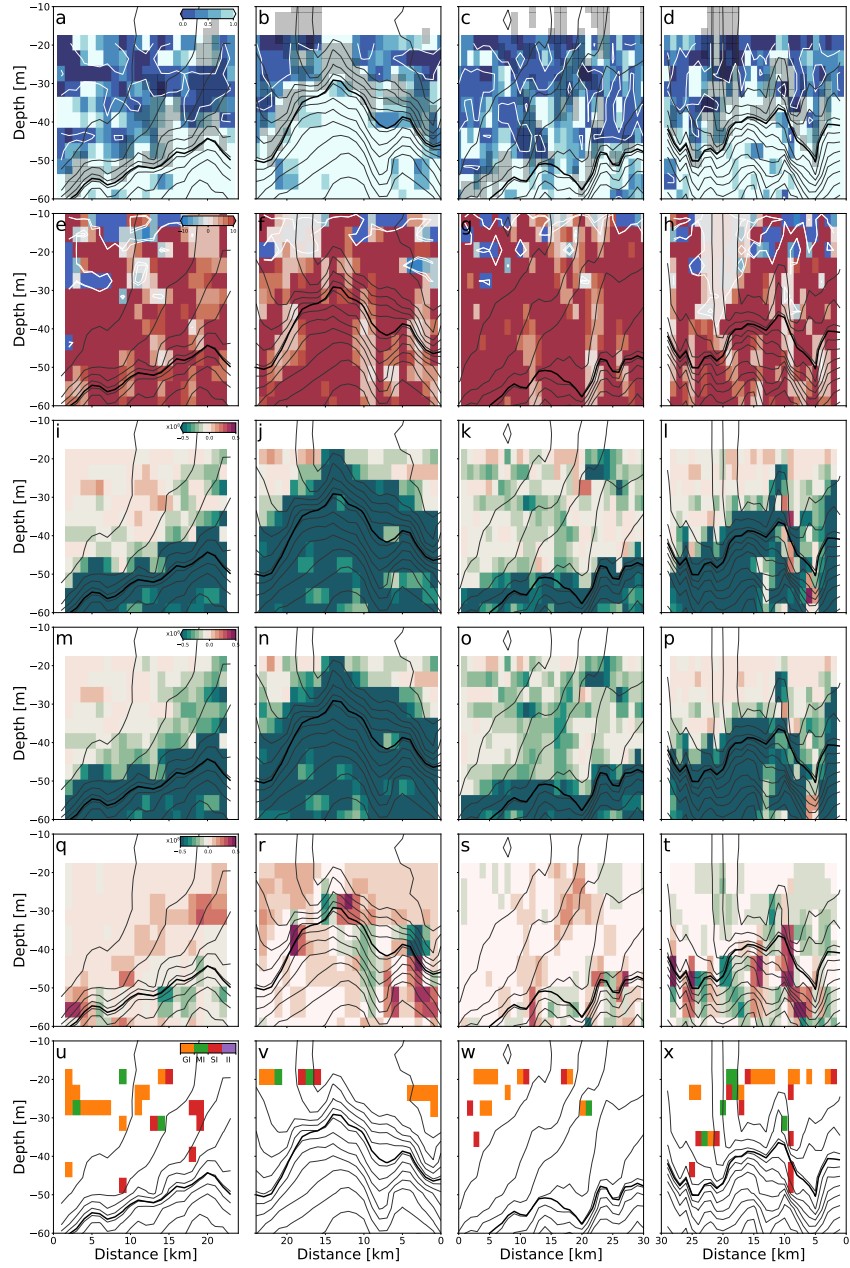

**Figure 8.** Vertical sections for the two Scanfish/ADCP transects that crossed the filament. From left to right, the columns correspond to the offshore south and north fronts and the nearshore south and north fronts (see Figure 1). a to d) gradient Richardson number ($Ri^G$), e to h) balanced Richardson number, $Ri^B$ , i to l) Ertel Potential Vorticity (EPV), m to p) vertical component of EPV, q to t) baroclinic component of EPV, u to x) Submesoscale instability analysis. The thin black contour lines show density (1024.2-1027.2 kg m$^{-3}$ in 0.05 kg m$^{-3}$ increments) and the thick black contour lines mark the depth of the surface mixed layer (SML) defined by a density change of 0.1 kg m$^{-3}$. The white contour lines in (a) to (d) is for 0.25, and in (e) to (h) for -1 and 1. The transects have been cropped (width and depth) to focus on the filament fronts and the SML and thus are of different length depending on available data. The light grey shading in (a) to (d) indicates estimated horizontal buoyancy gradients that exceed $10^{-7}$ s$^{-2}$.



Within and below the pycnocline $Ri^G$ exceeded 1 indicating stable conditions (Figure 8a to 8d). Within both filament cores and southern fronts and in the northern front of the nearshore transect, $Ri^G$ was consistently $< 0.5$ and even $< 0.25$ within the SML. Although variable, these $Ri^G$ values extended over the full SML and, in the southern fronts, may follow the sloping isopycnals. $Ri^G$ appears to exceed 1 just north of the offshore northern front (Figure 8b). Baroclinic instabilities can occur for all Ri, a value of $Ri^G < 1$ indicates that the flow may be susceptible to symmetric instabilities, while a value of $Ri^G < 0.25$ is generally interpreted to mean that the velocity shear is able to overcome the stratification, leading to turbulence and thus mixing through Kelvin-Helmholtz instabilities.

To delve further into the susceptibility of the filament fronts to instabilities, the Ertel potential vorticity (EPV) was calculated to see if conditions favoured submesoscale instabilities. Generally, instabilities may form where EPV is the opposite sign of $f$ (Thomas et al., 2013); i.e., positive in the southern hemisphere where $f$ is negative. Using ship-based measurements (ADCP and Scanfish) an observational approximation of EPV could be calculated, assuming gradients across the front are much greater than along the front (e.g., Thompson et al., 2016; Adams et al., 2017):

$$q = (f - \partial v \ \partial x^{-1})N^2 + \partial v \ \partial z^{-1} \ M^2 \tag{2}$$

where $q_{vert} = (f - \partial v \ \partial x^{-1})N^2$ is referred to as the vertical component of EPV and $q_{bcl} = \partial v \ \partial z^{-1} \ M^2$ the baroclinic component, and $M^2 = (-g \ \rho_\theta^{-1})(\partial \rho \ \partial x^{-1})$. In Figure 8i to l we plot the total EPV to identify possible regions where it is positive that could be prone to submesoscale instabilities. EPV was negative below the SML, identifying stable conditions, not susceptible to the formation of instabilities (Figure 8i to l). The lowest EPV values, at the base of the SML, correlate with high vertical stratification, indicating that it is this variable that mainly determines EPV values in this area. Patches of positive EPV were found throughout the SML of both fronts in both transects. The largest patches of positive EPV were found on the outer edge of the fronts, just beyond the strongest horizontal gradients. Furthermore, positive EPV values tended to be found in the upper SML within the fronts and outside the filaments, but within the lower SML in the filament core.

To connect to the Rossby number, we can express the vertical component of the EPV with Ro and the instability criterion $fq < 0$ becomes

$$f^2(1 + Ro)N^2 + f\partial v \ \partial z^{-1} \ M^2 < 0 \tag{3}$$

If we assume the flow to be mainly in geostrophic balance, $M^2 = \partial_y b = -f\partial_z v$, and with $N^2 > 0$ and $f \neq 0$ we can divide the instability criterion by $f^2 N^2$ and arrive at a non-dimensional criterion (Buckingham et al., 2019), now expressed with the balanced Rossby and balanced Richardson number $Ri^B = N^2 \ f^2 \ M^{-4}$, for which the velocities are replaced by the geostrophic velocities

$$(1 + Ro^B) - \frac{1}{Ri^B} < 0. \tag{4}$$



where $Ro^B$ is the balanced Rossby number, defined as the ratio of the geostrophic relative vorticity and the Coriolis force (Figure 8e-h). From the instability criteria we can see that the second baroclinic term contributes to negative $fq$ if $Ri^B$ is positive and is hence mostly destabilizing and more so for small (balance) Richardson numbers, while the contribution from the first (vertical) term depends on the sign and magnitude of the Rossby number. For sufficiently negative $Ro^B < -1$, i.e. for anticyclonic motion (anti-clockwise in SH), the latter will contribute to negative $fq$ while cyclonic motion will tend to stabilize.

The baroclinic EPV component (Figure 8q-t) was almost always positive, thus contributing to destabilization as expected. Positive EPV mainly occurred when the vertical component approached zero or was positive. In areas with $|Ro| < 1$, which was almost always the case in our Rossby number observations, it is the magnitude of the vertical stratification that determined if EPV will be positive (low $N^2$) or negative (high $N^2$). Considering the southern front, $N^2$ was consistently lower in the offshore transect (Figure 3c) than the nearshore (Figure 3d), and as a result the EPV is more consistently positive in the former. This result indicates that the southern front of the offshore transect is more susceptible to submesoscale instabilities, and the nearshore transect has already become more stable as it slumps (sloping isopycnals) and undergoes restratification (higher $N^2$).

Where EPV is positive, the magnitude of the balanced Richardson number can identify regions susceptible to gravitational ($Ri^B < -1$), mixed gravitational-symmetric ($-1 < Ri^B < 0$), symmetric ($Ro^B < 0$: $0 < Ri^B < 1$; $Ro^B > 0$: $0 < Ri^B < Ro^{B\,-1}$), and inertial ($Ro^B < 0$: $1 < Ri^B < Ro^{B\,-1}$) instabilities (e.g., Thomas et al., 2013, 2008; Thompson et al., 2016; Adams et al., 2017). Stable conditions occur when $Ri^B > Ro^{B\,-1}$.

The upper SML on either side of the filament was mainly susceptible to gravitational instabilities (Figure 8u to x). Mixed and symmetric instabilities were found within the fronts near the strongest horizontal gradients, and scattered along the bottom of the SML. Including data where EPV is negative, but near zero, showed mixed and symmetric instabilities occurred consistently within the fronts, over the full SML (not shown). The exception was in the southern front of the nearshore, where the occurrence of instabilities was more scattered.

## 5   Discussion and Summary

A multi-platform approach combining remote-sensing data (SST, SSH), in-situ measurements (cross-sections of filament) and drifter-data (probing along-filament and in the upwelling region) allowed a comprehensive analyses of a major filament encountered during November/December 2016 in the Benguela upwelling region off the coast of Luederitz, Namibia. The filament formed in the second week of November 2016 through the interaction of mesoscale cyclonic and anticyclonic eddies, which pulled cold and fresh, upwelled water offshore. The filament was about $2°$ C colder, 0.3 g kg$^{-1}$ fresher and 0.3 kg m$^{-3}$ denser compared to the ambient water, and had a width of about 30-50 km, bounded by temperature gradients of about $0.03°$ C / km. We observed a double-core structure in the onshore Scanfish transect. The length of the filament (about 400 km) was above average for this region (Hösen et al. (2016)). Also based on SST images Hösen et al. (2016) reported that most filaments persist for only 2-12 days, and only rarely do they reach 48 days. Drifter pathways from a depth of 15 m revealed that the filament was still present 40 days after the start of the cruise while the surface SST signature was already lost. This suggests




using SST to predict filament occurrence and lifetime likely leads to underestimates of both. In the vertical, the filament was characterized by a surface mixed layer (SML) less than 40 m deep, 20 m shallower than outside of the filament. At the bottom of the SML was a strongly stratified pycnocline, that was thinner on the southern side of the filament (<10 m) than the northern side (> 20m), and thinnest and weakest within the filament.

The largest horizontal density gradients were present in the mixed layer on the north side of the filament as well as in the middle of the filament below 20 m depth due to the doming of isopycnals from the upwelled cold waters. The salinity signature showed fresh water being pushed southward across the southern edge of the filament. This is consistent with Ekman transport driven by down-front winds, i.e. winds toward the north-west. Outside of the filament there are compensating lateral gradients in temperature and salinity resulting in no density gradient. In spite of the largest horizontal density gradients occurring toward the northern side of the filament, the largest velocities were found on the southern side of the filament and in the middle of the filament for the offshore scanfisch transect. Those larger velocities can be explained with the presence of the counter-rotating eddies that resulted in strong westward velocities in-between them at the southern side of the filament.

Submesoscale motions arise through surface frontogenesis that develops along the edges of the cold filament, as it is pulled offshore and strained between the mesoscale eddies. Associated with these submesoscale motions are relatively flat horizontal kinetic energy (KE) spectra for scales < 15 km followed by a steepening of the spectral slope at wavelengths larger than 15 km where the rotational flow component, derived from Helmholtz decomposition, becomes dominant. The divergent component of KE decreased sharply for scales > about 15 km while divergent and rotational components contributed equally for smaller scales. This is consistent with the cross-track KE component dominating for scales > 15 km, since for horizontally non-divergent flow $E_t = nE_l$ (where $E_t, E_l$ are across-track and along-track KE components respectively and with $E \propto k_h^{-n}$ the KE spectral power law) and hence $E_t > E_l$ for $n > 1$ (Charney, 1971). The transition in slope suggests that unbalanced flow becomes important at wavelengths below 15 km, which corresponds roughly with the first internal Rossby Radius of deformation in the region (e.g. Chelton et al., 1998).

The transition of 15 km is relatively small compared e.g. to the California upwelling region, which has lower mesoscale energy levels than the Benguela upwelling region, the latter beeing influenced by strong Agulhas eddies. For the California upwelling region Chereskin et al. (2019) report a transition scale of about 70 km for latitudes of about 30° - 35° N (slightly further poleward than our study area). Rocha et al. (2016) found a transition scale of about 40 km for Drake Passage, and identified inertia gravity waves to be dominate for scales < 40 km. Callies and Ferrari (2013) found steep spectra for scales 20-200 km for the Gulf Stream region, which is very similar to the spectra we found here. In contrast they found flatter spectra (inconsistent with interior QG) for the interior North Pacific which has little eddy activity. Hence our findings corroborate that a smaller transition scale is an indication of high mesoscale kinetic energy (Chereskin et al., 2019; Qiu et al., 2018).

We found spectral slopes do not change with depth (not shown), which was also found by Rocha et al. (2016), and is not compatible with SurfaceQG turbulence that predicts a steepening of spectra at deeper depths. The spectral slope close to $-5/3$ or $-2$ for scales 5-15 km may indicate inertial gravity waves, stratified turbulence or other mixed layer instabilities, that project on the small scales and flatten the spectra. We found evidence for Kelvin-Helmholtz and symmetric instabilities that may contribute to this forward cascade.



Through measurements from different platforms, we could explore different scales and thus captured a range of order of magnitudes for the Rossby number Ro. Because the two considered drifter releases explored nearby areas nearly distinctively,

we could analyze spatial and temporal Ro-variations of the filament as compared to its environment quasi separately and simultaneously. We applied a scale-dependent analysis of Rossby number to the two drifter releases using $2^{nd}$ order velocity structure functions and found that |Ro| inferred from the drifter trajectories of the southern release (which remains trapped in the filament) reached 0.5 at a scale of 3 km, and exceeded 1 for scales smaller than 0.5 km, pointing to the presence of ageostrophic submesoscale dynamics within the filament, because the contributions to the small separation regime are mainly

from the initial period immediately after the drifter deployments in the filaments. In contrast, for the northern release (which followed many different currents across a larger region outside of the filament) the method revealed that |Ro| reached 0.5 at a scale of 1 km, but never reached 1. While for the southern release for scale < 15 km |Ro| is twice that of the northern release, at larger separation distances (> 15 km), mainly corresponding to periods after the drifters left the filament, the differences between the Ro of the two releases diminish and yield |Ro| $\ll$ 1, consistent with the mesoscale dynamics of the open ocean.

Timeseries of Ro estimated from the ratio of relative vorticity and f were derived by applying methods of Molinari and Kirwan (1975) to triplets of drifters both inside and outside of the filament revealing horizontal variation of the Rossby number. |Ro| of $\mathcal{O}(0.1)$, both deduced from the satellite altimetry as well as from the drifter triplets out of the filament, were linked with mesoscale eddies. The time dependent |Ro| from this triangle approach showed $\mathcal{O}(0.5)$ values with occasional values > 1. The drifters' Ro had a positive bias, which means that they predominantly sampled inside of the filament on the cyclonic

side of the fronts. The maps of Ro interpolated from the ADCP tracks and derived from SSH showed similar values and Ro polarities, while only the former had visible hotspots exceeding 0.5, but not 1. |Ro| was found to be > 1 in the individual transects using Scanfish and ADCP data. For these transects Ro was uniform in the vertical, but varied across the filament and its fronts. Higher values were associated with the filament core, where horizontal and vertical velocity shear were highest. High variability (changes in sign) were observed across both fronts.

Overall, our Rossby number analyses support the idea that there are hotspots of elevated |Ro| $\geq$ 0.5, that were linked with the filament, with |Ro| occasionally exceeding $\mathcal{O}(1)$, often cited as a criteria for submesoscales (e.g. Thomas and Ferrari, 2008). The drifter-derived scale dependent Rossby numbers for the two releases are similar for scales larger than about 10 km while the Rossby numbers inferred from the drifters in the filament start to increase significantly for scales smaller than 10 km. This finding is roughly consistent with the flattening of the KE spectra from the ship ADCP data around a scale of about 15 km.

Our scale dependent Rossby number reminds to what Callies et al. (2015) observed from spectral estimates in the Oleander and LatMix summer experiments in the western subtropical North Atlantic. They concluded that their observed submesoscale flows follow quasi-geostrophic regimes with Rossby numbers < 0.5 and that only for scales < 1km they expected the flows to become strongly ageostrophic with Rossby numbers reaching 1. In fact, our KE spectra from the Benguela upwelling region are obtained for Southern Hemisphere spring and are similar to their summer spectral slopes of $k^{-3}$ for scales > 15 km. Their

spectral slopes change to a frontal $k^{-2}$ in winter for those and hence Callies et al. (2015) conclude that in winter, mixed layer baroclinic instabilities inject energy around scales between 1-10 km, driving an inverse cascade of kinetic energy that energizes



the mesoscale, while the summertime flow is dominated by quasi-geostrophic turbulence with a weak forward energy cascade for scales < 15 km.

Our finding of flatter spectral slopes and a moderate increase in Rossby number at scales below 15 km, particularly within the filament, points to the existence of a forward energy cascade at those scales and the question was whether evidence of processes contributing to the cascade could be found. Within the sloping isopycnals of the southern filament front positive $Ri^B$ ($> 1$) indicate ageostrophic baroclinic instabilities and low $Ri^G$ suggest these regions were susceptible to Kelvin-Helmholtz ($< 0.25$) and symmetric instabilities ($< 1$). Furthermore, south of the filament along the nearshore transect, the combination of high $Ri^B$ values, relatively high $N^2$ values (e.g., relative to the offshore transect) and compensating lateral density gradients suggest restratification has occurred through frontal slumping and surface-layer instabilities, followed by mixing (Timmermans et al., 2012). Positive EPV values —and indication of submesoscale instabilities —tended to be found in the upper SML within the fronts and outside the filaments, but also in the filament core in the lower SML. The southern front of the offshore transect is more susceptible to submesoscale instabilities, and the nearshore transect has already become more stable as it slumps (sloping isopycnals) and undergoes restratification (higher $N^2$). Evidence for turbulent mixing associated with submesoscale instabilities was recently found by Peng et al. (2020) in a filament closer to the upwelling region and south of the filament used in this study. Peng et al. (2021), who also involved in their analysis two drifter triplets deployed in a more mature phase of the filament, also found moderate Rossby numbers of O(1) and that the magnitude of the stabilizing vertical vorticity is too small to compensate for the destabilizing baroclinic effect in the central region of the front thereby indicating that the conditions for symmetric instability are fullfilled. Consistent with our findings, this suggests that indeed while scales > 15 km are dominated by quasi-geostrophic dynamics, the flatter spectral slopes for scales < 15 km are associated with a forward energy cascade, initiated by gravitational and symmetric instabilities leading to turbulence and dissipation.

*Data availability.* Drifter data are available through the Global Drifter Progam (http://www.aoml.noaa.gov/ phod/dac/index.php). SSH data were downloaded from https://data.marine.copernicus.eu/. Code, ship ADCP and Scanfish data are available here: https://github.com/rpnorth/Benguela

*Author contributions.* R. North programmed the analyses Code, did the analyses of ADCP, Scanfish data and wrote much of the text, J. Dräger-Dietel analyzed the drifter data, wrote text and contributed to editing and interpretation, A. Griesel wrote text, contributed to interpreting the results and editing the paper and aquired the funding.

*Competing interests.* No competing interests are present.





*Acknowledgements.* We thank the science party and the crew of cruise M132 on R/V Meteor for their support and Kerstin Jochumsen as lead scientist. This paper is a contribution to the project L3 (Meso- to submesoscale turbulence in the ocean) of the Collaborative Research 540 Centre TRR 181 "Energy Transfer in Atmosphere and Ocean" funded by the German Research Foundation (DFG).



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
