# Peer review of "Characterization of physical properties of a coastal upwelling filament with evidence of enhanced submesoscale activity and"

_EGUsphere, 2023_

## Referee Comment (RC2)

**EGUSPHERE-2023-1051**

**CHARACTERIZATION OF PHYSICAL PROPERTIES OF A COASTAL UPWELLING FILAMENT WITH EVIDENCE OF ENHANCED SUBMESOSCALE ACTIVITY AND TRANSITION FROM BALANCED TO UNBALANCED MOTIONS IN THE BENGUELA UPWELLING REGION**

REFEREE REPORT

**1. General comments**

This paper deals with results from a hydrographic survey of an upwelling filament in the Benguela Upwelling System. The objective of the paper is to investigate the evolution, lifetime and physical characteristics of the surveyed filament. Special emphasis is made on detecting the presence of unbalanced, ageostrophic motions. The paper presents ADCP, Scanfish, remote sensing and drifter measurements that were used observe the hydrography and multi-scale dynamics of the filament and adjacent circulation. My opinion is that this is a timely contribution to the literature on upwelling filaments. However, i find the manuscript with shortcomings that must be addressed before publication. Therefore I am recommending that the manuscript be sent back to the authors for **minor revisions**.

My main concern is that the manuscript is written and organised in a way that doesn't flesh out its main conclusions. The analysis part of the manuscript (section 4), that deals with the Rossby-number analysis should be made the main part of the results presentation. The results that precede it (section 3), although important on their own, should be made secondary and only those pertinent to section 4 should be in the main body of the manuscript. The remainder may be moved to a supplementary information part or removed altogether.

Secondly, the structure of the filament is discussed in several sections in relation to its hydrographic and dynamic properties, however it is difficult to maintain a visually coherent picture of the processes being discussed and the filament structure. Therefore I recommend the authors to make a diagram of the filament cross-section that shows the different processes that the authors identify, especially those that occur differently on the filaments northern and southern boundary and over its depth.

**2. Specific comments**

2.1. **Manuscript text.**

- Introduction (lines 37–39). This sentence about climate models appears out of place. Either remove it or develop it further.

- Introduction (lines 83–84). This sentence is awkward, as it appears to move the focus of the paper from the filament itself to the wider Benguela upwelling region. It is better to remove it.
- Submesoscale instabilities (line 407). The assumption of a flow (which? the one associated to the filament?) mainly in geostrophic balance needs to be justified.
- Discussion and summary. The lack of references to figures makes it hard to follow the discussion.

2.2. **Figures.** In general, figures need to be improved for readability and quality.
- Subfigure labels size needs to be increased.
- Figure 1. This figure is loaded with information, but it is confusing as the elements of the figure lack any reference. I would do, if possible, some or all of the following:
    - Add the start and end points of the cruise and the dates.
    - Add the dates of the transects used to determine the KE spectra.
    - Add the dates of the Scanfish transects.
  and fix the figure boundary and the colorbar.
- Figure 5. Add a legend to panel a.
- Figure 6. I would remove the ship's track (grey broken line).
- Figure 7. In the label of the x-axis should there no be a "log"?

---

## Author Comment (AC1)

**Reply to Reviewer 1:**

This reviewer has not specifically requested any explicit changes to the article. We will reply below to the reviewer's comments, which are more general and descriptive.

**Reviewer 1**

1. The Rossby number is not small, but there is no evident secondary circulation. Kinetic energy wavenumber spectra are calculated from four long ship tracks. A fuss is made over the spectra being somewhat steeper at smaller wavenumbers and less steep at larger ones (down to a wavelength of 200 m). This is consistent with previous experience. The horizontal velocity is mostly rotational at smaller wavenumbers, consistent with geostrophic balance, and nearly equipartitioned with its divergent component at larger wavenumbers, the latter consistent with partly ageostrophic submeoscale motions or a modest presence of inertia-gravity waves.

   **We agree that a transition is expected at some scale, when the rotational part dominates. The point we made was that the transition scale is small compared to the California upwelling region, which can be explained by the higher eddy activity. Because slope changes in wavenumber spectra of total kinetic energy are not easily identifiable in general see e.g. also previous studies (e.g. Rocha et al., Chereskin et al.), we were able to shed more light on the transition-scale with the Helmholtz decomposition and the distinction between along/cross track components.**

2. The stated context is to identify submesoscale instabilities of several types. In my view this is not very conclusive. Patchy values of possible exceedance of stability thresholds are found, especially in the surface boundary layer where we can expect small-scale turbulence to be active, but otherwise there seems to be no systematic pattern. The authors claim stronger confirmation of instabilities than I think is justified.

   **Our aim with the multi-platform approach was to connect the mesoscale processes with the submesoscales. We agree that the patchiness of the instability results might not necessarily be conclusive and that enhanced horizontal resolution and repeated sections in time would be preferable. We added in the Conclusions: "The EPV and instability diagnostics are sensitive to the resolution of the data. Likely, with higher horizontal resolution, density gradients and hence positive EPV would be enhanced, where it is barely positive with the resolution of the Scanfish. The patchiness of the instability analysis results illustrates the challenges associated with connecting mesoscale with smaller scale processes and enhanced horizontal resolution and repeat transects in time are preferable in future missions"**

---

## Author Comment (AC2)

**Reply to Reviewer 2:**

My main concern is that the manuscript is written and organised in a way that doesn't flesh out its main conclusions. The analysis part of the manuscript (section 4), that deals with the Rossby-number analysis should be made the main part of the results presentation. The results that precede it (section 3), although important on their own, should be made secondary and only those pertinent to section 4 should be in the main body of the manuscript. The remainder may be moved to a supplementary information part or removed altogether.

**To make the manuscript better readable and flesh out the results we restructured the manuscript in the following way: We put the old section (former 3.1) with old Figure 2 in the supplementary material and suspended former section 3. After a short general introduction of the filament at the beginning of the "Data and methods" section (2) now the results section 3 follows starting with the section about the spectra (3.1) and about the regional near surface (3.2) and the Scale dependent Rossby number analysis (3.3) prominently as the first part of the results. We feel that the description and analysis of the velocity and density structure of the filament as evident in the onshore and offshore scanfish sections is important as only with them one can understand where the enhanced Rossby numbers and flatter spectral slopes stem from and also how properties change from onshore to offshore. We therefore kept it (and the corresponding figures) in the main body but it appears now later as an introductory section (3.4) for the two last sections 3.5 (Cross-filament variability in Ro) and 3.6 (Submesoscale instabilities)**

Secondly, the structure of the filament is discussed in several sections in relation to its hydrographic and dynamic properties, however it is difficult to maintain a visually coherent picture of the processes being discussed and the filament structure. Therefore I recommend the authors to make a diagram of the filament cross-section that shows the different processes that the authors identify, especially those that occur differently on the filaments northern and southern boundary and over its depth.

**We have created a new schematic (Fig. 8) that summarizes the filament processes and hydrographic properties and discuss it in the conclusions.**

Figures. In general, figures need to be improved for readability and quality.

**We have increased font sizes in all figures, moved color bars for better readability (Fig.1,3,5,6), and changed color bars and color bar limits for better readability (Fig. 3,5,6).**

**Response to specific comments**

1. Introduction (lines 37–39). This sentence about climate models appears out of place. Either remove it or develop it further
   **We removed the sentence: Concerning climate models a better understanding of their meso- and submesoscale processes is needed in order to improve the parameterisations of their small scale dynamics.**

2. Introduction (lines 83–84). This sentence is awkward, as it appears to move the focus of the paper from the filament itself to the wider Benguela upwelling region. It is better to remove it.

**We agree that this sentence was formulated in a misleading way, but has importance since one key result of this article is that the transition scale from balanced to unbalanced flow appears to be smaller than in other upwelling regions with less mesoscale energy. We rewrote the sentence such that it has its focus on the filament rather than the upwelling region: "Here, we consider whether the filaments in the Benguela upwelling region, that are influenced by the passage of strong Agulhas eddies, exhibit spectral characteristics that differ from the filaments in the California upwelling region with its lower eddy kinetic energy".**

3. Submesoscale instabilities (line 407). The assumption of a flow (which? the one associated to the filament?) mainly in geostrophic balance needs to be justified.
   **We reformulated this, because the word 'assumption' was not appropriate since we do not assume a priori that the flow is in geostrophic balance. We note now that the flow associated with the filament has ageostrophic components but that the submesoscale instability analysis is based on the balanced Richardson and Rossby numbers (Thomas et al. 2013).**

4. Discussion and summary. The lack of references to figures makes it hard to follow the discussion.
   **We included references to the Figures where appropriate. Also the new schematic (Fig. 8) should make it easier to follow the discussion.**

5. Subfigure labels size needs to be increased.
   **We increased the label sizes.**

6. Figure 1. This figure is loaded with information, but it is confusing as the elements of the figure lack any reference. I would do, if possible, some or all of the following: – Add the start and end points of the cruise and the dates.
   – Add the dates of the transects used to determine the KE spectra.
   – Add the dates of the Scanfish transects.
   and fix the figure boundary and the colorbar.
   **We added the dates for the Scanfish transects and start date of the cruise in the figure. We added numbers in the figure to the long ship transects used for the spectra and put the corresponding dates in the figure caption. We also fixed the figure boundary and colorbar.**

7. Figure 5. Add a legend to panel a. **We added a legend.**

8. Figure 6. I would remove the ship's track (grey broken line) **We improved the color scales in this figure and removed the shiptrack from a-c and e. However, we kept the shiptrack for panel d since it shows where the data for this figure comes from.**

9. Figure 7. In the label of the x-axis should there no be a "log"? **No: the x-axis is already logarithmically.**